# A Data Driven Approach to Identify Safe and Adequate Schemes for Vitamin D Fortification

**DOI:** 10.3390/foods11243981

**Published:** 2022-12-08

**Authors:** Tue Christensen, Gitte Ravn-Haren, Rikke Andersen

**Affiliations:** 1Research Group for Nutrition, Sustainability and Health Promotion, National Food Institute, Technical University of Demark, 2800 Kongens Lyngby, Denmark; 2Research Group for Risk Benefit, National Food Institute, Technical University of Demark, 2800 Kongens Lyngby, Denmark

**Keywords:** fortification, vitamin D, food, dietary-intake data, optimization tool, fortification strategy, linear modelling

## Abstract

Food fortification is a strategy to increase low vitamin D intake. In order to avoid the intake of a population exceeding the upper tolerable intake level, the right choice of food groups to fortify is of crucial importance. An automated fortification tool was developed based on dietary intake data from the Danish National Survey of Dietary Habits and Physical Activity 2011–2013 (DANSDA), taking into account the energy contribution of the fortified food. The fortification of food group is a variant in the linear modelling, where the optimization ensures the lowest possible variation in deviation of the calculated intake and the target intake. The resulting tool demonstrated that the lowest limit of fortification, where the model works, is 12 µg/10 MJ, when fortification of any food group is allowed. The tool also demonstrated that, by increasing the allowed upper level of fortification from 12 µg/10 MJ up to 30 µg/10 MJ, the food groups selected for fortification and the level of fortification in those food groups may change. Specifically, fewer food groups seem to be needed as the upper level of fortification is increased. The optimized scenarios, using the food groups, including milk, cheese, cereals, fats, and juice, were tested on dietary-survey data and demonstrated that all the projected scenarios manage to lift the median vitamin D intake to the targeted intake safely. A data-driven approach was used to develop a simple, fast, and automated fortification tool to test different vitamin D food fortification strategies.

## 1. Introduction

Vitamin D status plays an important role in bone health, and low vitamin D status is also linked to non-skeletal diseases and mortality [1,2]. The skin production of vitamin D through sun UVB-radiation can be limited by personal factors (e.g., skin pigmentation, genotype, age), lifestyle factors (e.g., sunscreen, sun avoidance), and geographical factors, such as living at Northern latitudes, where the UVB radiation is too low for skin production during wintertime (October to March) [3,4]. Therefore, oral intake of vitamin D is important to maintain adequate vitamin D status (measured as serum 25-hydroxyvitamin D (25 OHD)).

Three main strategies can increase oral vitamin D intake: (1) habitual diet, (2) supplementation, and (3) food fortification. All strategies have advantages and disadvantages. Habitual diet contains other beneficial nutrients, but a limited number of foods (especially fish) naturally contain vitamin D, and the intake from habitual diet is generally low [5]. Therefore, regular information campaigns are needed in order to increase, for example, fish intake. Vitamin D containing supplements increases 25 OHD concentration, as long as the supplements are taken regularly across all age and socio-demographical groups, which is seldom the case [6]. Furthermore, the use of high-dose supplements increases the risk of exceeding the tolerable upper intake level (UL) [7]. Food fortification can be a safe and sufficient strategy to increase vitamin D intake in all population groups, but it depends on the food or combination of foods fortified and the doses used. Vitamin D fortification of several different food groups instead of concentrating on only a few food groups has been suggested to successfully reach different population groups with dissimilar dietary habits [8]. Consumers’ awareness and perception of fortification might affect the strategy [9].

The fortification of foods can be introduced by the authorities as mandatory fortification, or it can be voluntary. Vitamin D-fortification policies differ between Nordic countries. Finland is an example of a successful voluntary-vitamin D-fortification policy that has contributed to improved vitamin D status in the general population [10]. In several member states in EU, fortification is permitted without pre-approval, but in Denmark, companies must apply for pre-approval [11]. Food fortification with vitamin D in Denmark is regulated by the food authorities and is voluntary. Allowed food groups are: beverages, milk & milk products, margarines and other fatty products, breakfast cereals, chocolate, bars and alike, ice-cream, crackers, and biscuits [12]. However, few vitamin D fortified products are on the Danish market. Besides, no published intake data are available on consumption of fortified foods in Denmark.

The effect of consuming vitamin D fortified foods on 25 OHD concentration has been demonstrated in several randomized-controlled trials [13,14]. A systematic review and meta-analysis including 23 studies found that food-vitamin D fortification is an effective strategy to increase 25 OHD concentration, although the response to vitamin D fortified food consumption can be influenced by age, BMI, and baseline 25 OHD concentrations [15]. Fortifying a single food item (e.g., juice or milk) is a challenge similar to supplements, since it does not increase vitamin D intake or status in non- or low consumers [16]. Madsen et al. [13] found that vitamin D fortification of two foods (milk and bread) during six months reduces the decrease in 25 OHD concentrations during winter and ensures 25 OHD concentrations above 50 nmol/L in children and adults in Denmark. Grønborg et al. [14] found that vitamin D fortification of four different foods for 12 weeks during winter was effective in increasing 25 OHD concentration and reducing the prevalence of very low vitamin D status among women of Danish and Pakistani origin.

The main aim of the present paper was to develop a simple, fast, and automated fortification tool to test different vitamin D-food-fortification strategies using population-based dietary-intake data from a representative national dietary survey and taking into account the energy contribution of the fortified food groups. Dietary vitamin D intake from habitual diet (without supplements) and fortifiable foods in different combinations will be used as an example, and the resulting optimal fortification of food groups will be evaluated by the ability to raise median intake to Recommended Intake (RI) [2] and keeping the P95 below the tolerable Upper intake Level (UL) [7].

## 2. Materials and Methods

Dietary-intake data from the Danish National Survey of Dietary Habits and Physical Activity 2011–2013 (DANSDA) are expressed as individual food intake for men and women aged 4–75 in the years 2011–2013 [5]. DANSDA is based on 7-day dietary recordings, a recipe-collection, and food-composition data (FRIDA 2022) [17]. The dietary intake data is interpreted as food-composition-table items. Each food-composition-table item is assigned to a food group. Using individual food intake, food-grouped intake of the total weight in grams, energy, and vitamin D for each individual is calculated. This data are further aggregated into descriptive (mean, median) statistics, grouped according to age and sex. The median intake for food groups, age, and sex is used in the model.

The calculations on the dietary survey are used for an optimization model based on a linear-simplex model. The optimization model is constructed in an Excel-spreadsheet and optimization is obtained using Excel’s problem solver functions [18].

The rationale behind the optimization model was:A maximum of fortification per 10 MJ for all food groups can be set. As the recommended energy intake is roughly 10 MJ for most population groups, the upper limit pr 10 MJ secures a safe vitamin D intake even if individuals only eat fortified foods.The calculation of contributions at any given combination of sex, age, and food group will be based on the median energy intake and median food group intake for the data point.Any population group has a median intake of the nutrient subject for fortification.Any population group has a target for intake of the nutrient subject for fortification.

The optimization model has the fortification of food group as a variant and the outcome of the optimization is the lowest possible variation in deviation of the calculated intake, sum of effect of fortifications of food groups and background intake, and the target intake (RI) for the population’s sex and age groups. This is a linear model, so linear-simplex optimization can be deployed.

To explore how the tool performs with many food groups available, it is first tested with all available food groups. The lowest level of fortification is expressed as fortificant per 10 MJ using all food groups (“Milk and milk products”, ”Cheese and cheese products”, ”Ice cream, fruit ice and other edible ices”, ”Cereals and cereal products”, ”Vegetables and vegetable products”, ”Fruit and fruit products”, ”Meat and meat products”, ”Fish and fish products”, ”Poultry and poultry products”, ”Egg and egg products”, ”Fats, oils and their products”, ”Sugar, honey and products thereof”, ”Beverages”, ”Spices and other ingredients”, ”Other foods”, ”Potato and products thereof”, and ”Juice”), which is estimated by increasing the concentration from 10 µg/10 MJ until the optimizer can deliver a result. The scheme using all 15 food groups is also tested at 25 µg/10 MJ to see how the fortification scheme will change at altered conditions.

According to Article 4 of Regulation (EC) No 1925/2006 [19], non-fortifiable items are unprocessed foods, including but not limited to, fruit, vegetables, meat, poultry, and fish, and beverages containing more than 1.2% by volume of alcohol. Since European legislation does not allow just any food to be fortified, a series of optimal scenarios is calculated with a fortification level for vitamin D set at 20, 25, or 30 µg/10 MJ using only the following selected food groups: “Milk and milk products”, “Cheese and cheese products”, “Cereals and cereal products”, and “Fats, oils, and their products” with optional inclusion of “Juices”. The optional inclusion of juice is used to observe how the resulting fortification scheme will change if a food group is omitted. The scenarios used in the tool development are shown in Table 1.

The optimizers’ result for a given scenario is verified by calculating individual intakes for a complete dietary survey using fortified-food-composition items. In order to make the calculations, a conversion of the µg/10 MJ was converted to µg/100 g using the ratio between total energy and total intake in grams for the food groups on population level.

### Testing the Scenarios

The optimization schemes are tested by constructing a new food-composition table using the schemes to add fortification to the foods and recalculate the individual food intakes in the dietary survey with these new data. The individual intakes for the unfortified background intakes and the eight scenarios are then compared by medians and percentiles. The comparison of the scenarios is done in R [20]. Vitamin D intake is represented as µg/day. For all scenarios the median intake, 5th percentile (P5) and 95th percentile (P95) are calculated.

To compare the scenarios, boxplots are prepared with indentation indicating 95% confidence interval for the median. When there is no overlap of indentation between 2 boxes, the difference in median is statistically significant. The calculations for all scenarios are divided into age groups: 4–6 years, 7–10 years, 11–14 years, 15–17 years, 18–50 years, and 51–75 years, as well as into sex groups.

The UL used is 50 µg/day for smaller children aged 4–10 years and 100 µg/day for larger children aged 11–17 years as well as adults [7]. The UL is compared to P95.

Misreporting is commonly found in dietary surveys, and this misreporting may influence the mean of the intake. In addition, intake of certain food groups may be skewed. This method uses the median intake to minimize the effects of outliers and skewness in the intake data, as median gives a better central location under those circumstances.

## 3. Results

### Optimization Schemes

The food groups fortified in the model calculations and the dose of fortification are shown in Table 2. This table shows the optimized level of fortification for the food groups with the chosen fortifiable food groups taking into account different upper fortification levels. The fortification for the food groups is expressed as µg/10 MJ and µg/100 g. The model calculations showed that a minimum level of fortification with vitamin D is reached at 12 µg/10 MJ for a scenario where all fortifiable food groups can be fortified (scenario 1), and the tool assigned fortification to 15 out of 17 food groups. Increasing the level to 25 µg/10 MJ in scenario 2 allowed fortification of fewer food groups, i.e., nine out of the 17 food groups. For scenarios with restriction on allowed-food groups (scenarios 3–6), all allowed-food groups are chosen when upper level of fortification is below 30 µg/10 MJ. When the level is 30 µg/10 MJ, only three out of the five food groups are chosen for scenario 7, and two out of the four for scenario 8. Also, it appears that all scenarios having juice available use fortification of this near the maximal-allowed level. An Excel workbook with the optimizing tool is available at the Appendix A.

The calculated impact of the fortification on median intake using individual intake data from DANSDA is presented in Table 3. The target of fortification (reaching 7.5 or 10 µg vitamin D) is reached for all age groups for all scenarios, but when it is divided by age groups, some scenarios are below the target for girls in age group 11–17 years. This population group also has the lowest intake of vitamin D when foods are not fortified with vitamin D and has a lower food intake than most of the other age- and sex-groups. Scenario 4 (the regular scenario with least-food groups and lowest maximum for fortification within the food groups) has the lowest impact on the girls in age group 11–17 years. The scenarios with more-food groups or higher maximum level all perform better on lifting the median intake for girls in age group 11–17 years. None of the scenarios resulted in vitamin D intakes (P95) exceeding the UL for the investigated age or sex groups.

To compare the scenarios, boxplots are presented in Figure 1 and Figure 2. When only looking at age groups (Figure 1), it appears that all scenarios perform well and almost equally. If sex is taken into consideration (Figure 2), it is clear that, in all the fortification scenarios, the intake is increased more in males than in females, but also that the fortification goal of reaching a median value of 7.5 or 10 µg is reached for all age groups, except girls aged 11–17 years.

## 4. Discussion 

A simple, fast, and automated fortification tool to test different vitamin D- food-fortification strategies was developed using a data-driven approach and taking into account age and sex. The tool uses population based dietary-intake data from a representative national dietary survey [5] and it takes into account the energy contribution of the fortified food. Dietary vitamin D intake from habitual diet (without supplements) and fortifiable foods in different combinations were used, and the optimal combinations of foods were estimated by percentage of the population whose vitamin D intake was above RI and below UL. With this tool, it is possible to get an indication of how an optimal and safe fortification strategy can look like easily.

The automated-fortification tool demonstrated that the lowest limit of fortification, where the model works, is 12 µg/10 MJ, when fortification of any food group is allowed. The tool also demonstrated that, by increasing the allowed upper level of fortification from 12 µg/10 MJ up to 30 µg/10 MJ, the food groups selected for fortification and the level of fortification in those food groups may change. In particular, fewer food groups seem to be needed as the upper level of fortification is increased.

The method by Hirvonen et al. [8] was also based on fortification per energy unit and dietary-intake data, but it differs from our optimization tool by not letting the model automatically determine the optimal fortification of possible food groups. We tested our model on sex, children, and adults, while Hirvonen et al. [8] only tested on the adult population. Both models used RI and UL as lower and upper limits; however, RI can easily be replaced by AR in our model. Earlier models for food fortification focused on UL only [21,22]. Hirvonen et al. [8] calculated the non-fortified intake of vitamin D as the sum of the intakes from diet and dietary supplements. In our model, we did not include the intake from food supplements, as our focus was to develop a tool to test fortification strategies. The dietary intake can easily be replaced with the total intake in our model, thereby accounting for the contribution from food supplements. We assume that the Danish Food Administration might change the vitamin D food-supplement recommendations if mandatory fortification is introduced.

It is a strength that our model is based on median vitamin D and energy intakes from a representative sample of the Danish population including both children and adults. In our model, the calculation of contributions is based on the median energy intake and median food group intake for the data point instead of the mean values, as energy under-reporters as well as over-reporters are included in DANSDA [5]. Misreporters are excluded in Hirvonen et al. [8]. It is also a strength that our model is based on the actual dietary intake, which is easy, fast, and simple to use, and does not require advanced programs.

A limitation in our model (and in Hirvonen et al. [8]) is that the proportion of fortification is presumed to be 100% in each food group. This is not usually the case in real life. However, the model could be applied with a more detailed food grouping, pin-pointing the specific food groups that actually can be fortified. For instance, un-fortifiable unprocessed meat could be separated from fortifiable-processed-meat products. Another improvement of the model could be further progress from the linear-simplex model to a mixed-integer programming, since the latter will allow for the inclusion of more food groups into the model. Besides, dietary habits are always in transition. A simulation study [23] has demonstrated that a sufficient vitamin D intake is not possible without food fortification if carbon emission is to be within realistic limits. Unfortunately, we do not have Danish data taking into account a transition towards more sustainable food sources, but the tool can be used on new data as they appear.

In our model, the fortification per 10 MJ is converted to fortification per 100 g food by a factor calculated for each food group. It would be more correct to do this conversion per food, but in order to make fortification schemes that are simple to implement, we calculate the fortification in µg/100 g in the food group based on an average energy content per 100 g for the food group. Alternatively, the legislation should state fortification per 10 MJ for the actual food groups.

Juice was included in scenarios 3, 5, and 7 and not in scenarios 4, 6, and 8. The addition of juice as a fortifiable-food group increases the median vitamin D intake for the group that is most difficult to increase (girls aged 11–17 y). Including juice gives a more levelled intake for all population groups in terms of median intake and lower P95, compared to scenarios without juice.

Milk and milk products, and fat products are food groups that are often fortified with vitamin D, but our model shows that other food groups can be included in order to achieve a safe and sufficient strategy to increase vitamin D intake in all population groups.

In conclusion, a data-driven approach was used to develop a simple, fast, and automated fortification tool to test different vitamin D-food-fortification strategies. We assume this tool could be used with any food-consumption study capable of providing valid data on energy intake at food-group level and on current vitamin D intake or intake of other micronutrients.

## Figures and Tables

**Figure 1 foods-11-03981-f001:**
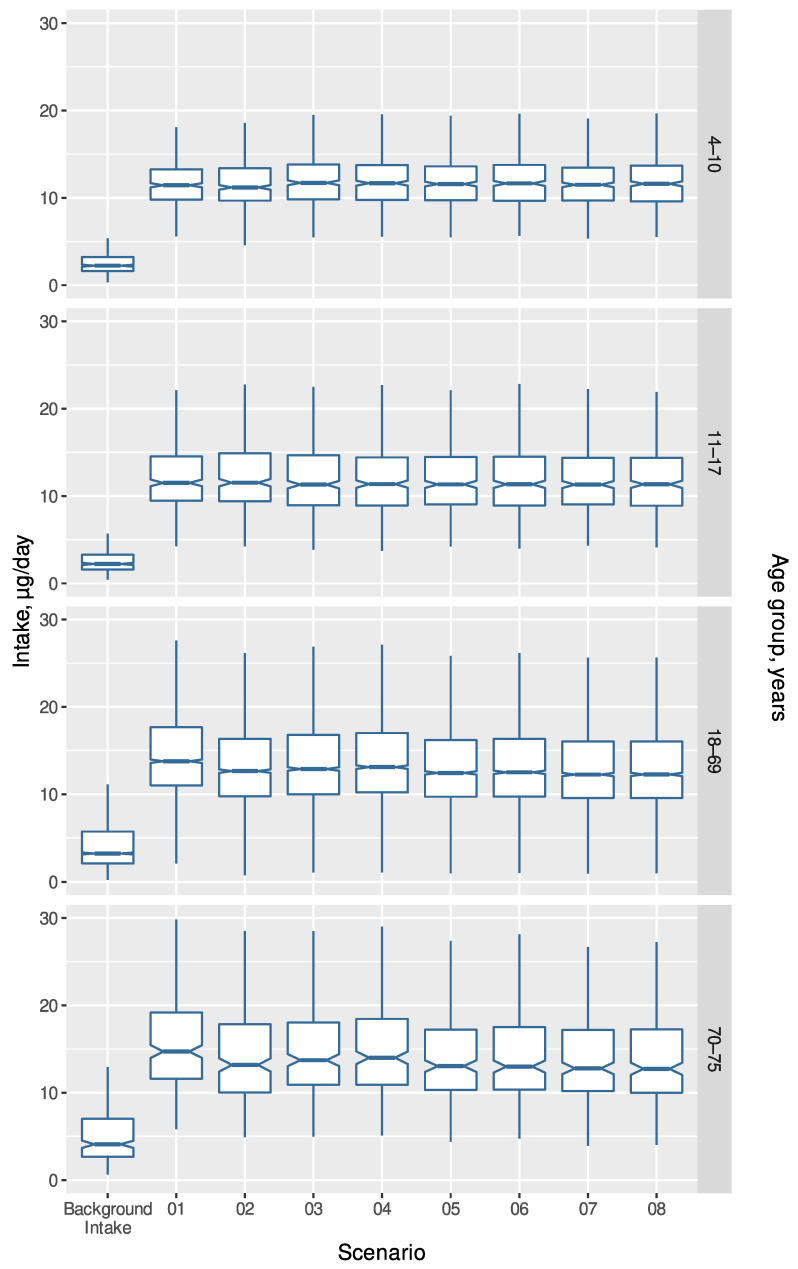
Calculated intake of vitamin D (µg/day) for background intake and the eight optimized scenarios by four age groups using individual food intakes from DANSDA. Scenarios are described in Table 1.

**Figure 2 foods-11-03981-f002:**
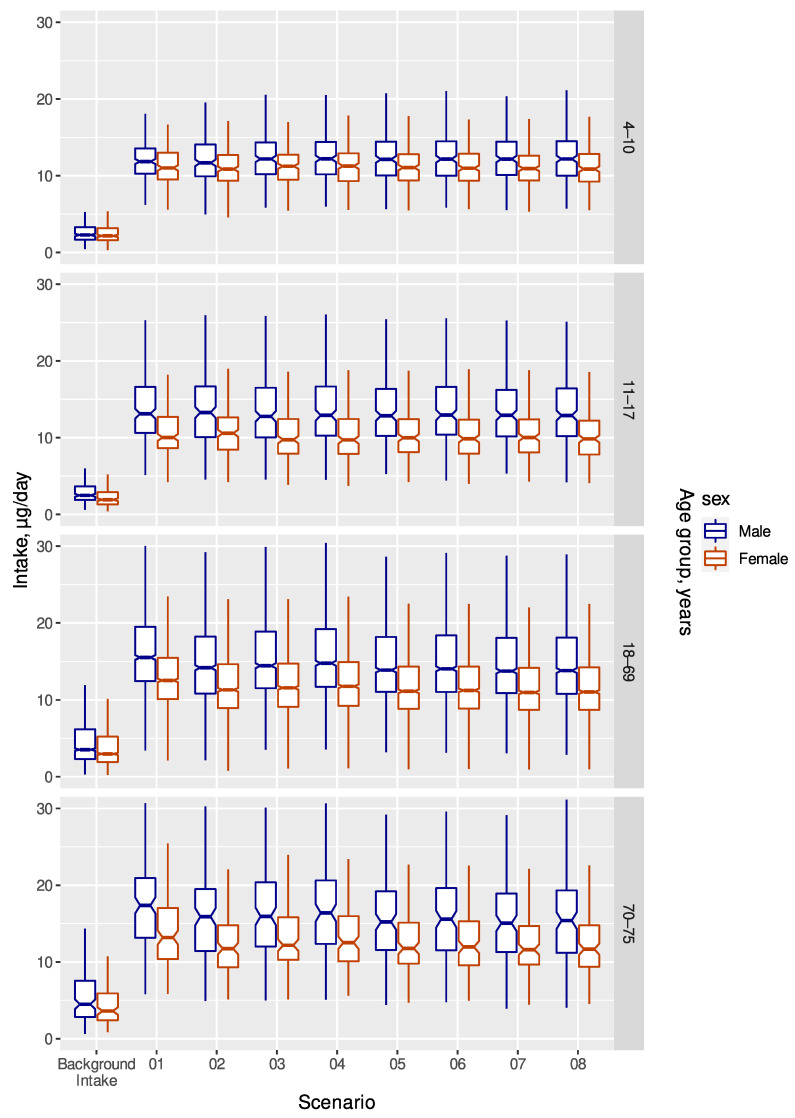
Calculated intake of vitamin D (µg/day) for background intake and the eight optimized scenarios by four age groups and sex using individual food intakes from DANSDA. Scenarios are described in Table 1.

**Table 1 foods-11-03981-t001:** Scenarios.

Scenario	Upper Level for Fortification, µg/10 MJ	Allowed Food Groups
Scenario 1	12 *	All (15) ^1^
Scenario 2	25	All (15) ^1^
Scenario 3	20	5 ^2^
Scenario 4	20	4 ^3^
Scenario 5	25	5 ^2^
Scenario 6	25	4 ^3^
Scenario 7	30	5 ^2^
Scenario 8	30	4 ^3^

* Lowest working value found for tool. ^1^ All food groups: “Milk and milk products”, ”Cheese and cheese products”, ”Ice cream, fruit ice and other edible ices”, ”Cereals and cereal products”, ”Vegetables and vegetable products”, ”Fruit and fruit products”, ”Meat and meat products”, ”Fish and fish products”, ”Poultry and poultry products”, ”Egg and egg products”, ”Fats, oils and their products”, ”Sugar, honey and products thereof”, ”Beverages”, ”Spices and other ingredients”, ”Other foods”, ”Potato and products thereof”, and ”Juice”; ^2^ Five food groups: “Milk and milk products”, ”Cheese and cheese products”, ”Cereals and cereal products”, ”Fats, oils and their products”, and ”Juice”, ^3^ Four food groups: “Milk and milk products”, ”Cheese and cheese products”, ”Cereals and cereal products”, and ”Fats, oils and their products”.

**Table 2 foods-11-03981-t002:** Optimized scenarios, level of fortification of food group per 10 MJ and per 100 g.

	Fortification	Scenarios							
Food Group	Unit	1	2	3	4	5	6	7	8
Milk and milk products	µg/10 MJ	2	25	20	20	25	25	24	25
	µg/100 g	0.274	0.571	0.457	0.457	0.571	0.571	0.548	0.571
Cheese and cheese products	µg/10 MJ	12	0	8	16	3	3	0	0
	µg/100 g	1.464	0.000	0.976	1.952	0.366	0.366	0.000	0.000
Ice cream, fruit ice, and other edible ices	µg/10 MJ	12	25	0	0	0	0	0	0
	µg/100 g	1.037	2.161	0.000	0.000	0.000	0.000	0.000	0.000
Cereals and cereal products	µg/10 MJ	12	15	20	20	25	25	27	28
	µg/100 g	1.418	1.773	2.364	2.364	2.955	2.955	3.191	3.309
Vegetables and veg. products	µg/10 MJ	12	0	0	0	0	0	0	0
	µg/100 g	0.435	0.000	0.000	0.000	0.000	0.000	0.000	0.000
Fruit and fruit products	µg/10 MJ	12	0	0	0	0	0	0	0
	µg/100 g	0.435	0.000	0.000	0.000	0.000	0.000	0.000	0.000
Meat and meat products	µg/10 MJ	12	0	0	0	0	0	0	0
	µg/100 g	1.055	0.000	0.000	0.000	0.000	0.000	0.000	0.000
Fish and fish products	µg/10 MJ	0	0	0	0	0	0	0	0
	µg/100 g	0.000	0.000	0.000	0.000	0.000	0.000	0.000	0.000
Poultry and poultry products	µg/10 MJ	12	25	0	0	0	0	0	0
	µg/100 g	0.733	1.528	0.000	0.000	0.000	0.000	0.000	0.000
Egg and egg products	µg/10 MJ	12	0	0	0	0	0	0	0
	µg/100 g	0.716	0.000	0.000	0.000	0.000	0.000	0.000	0.000
Fats, oils and their products	µg/10 MJ	12	0	20	20	4	7	0	0
	µg/100 g	3.508	0.000	5.847	5.847	1.169	2.047	0.000	0.000
Sugar, honey, and products thereof	µg/10 MJ	12	25	0	0	0	0	0	0
	µg/100 g	2.172	4.525	0.000	0.000	0.000	0.000	0.000	0.000
Beverages	µg/10 MJ	0	0	0	0	0	0	0	0
	µg/100 g	0.000	0.000	0.000	0.000	0.000	0.000	0.000	0.000
Spices and other ingredients	µg/10 MJ	11	19	0	0	0	0	0	0
	µg/100 g	0.091	0.157	0.000	0.000	0.000	0.000	0.000	0.000
Other foods	µg/10 MJ	12	25	0	0	0	0	0	0
	µg/100 g	2.717	5.660	0.000	0.000	0.000	0.000	0.000	0.000
Potato and products thereof	µg/10 MJ	12	24	0	0	0	0	0	0
	µg/100 g	0.469	0.938	0.000	0.000	0.000	0.000	0.000	0.000
Juice	µg/10 MJ	12	25	19	0	25	0	29	0
	µg/100 g	0.228	0.476	0.362	0.000	0.476	0.000	0.552	0.000

**Table 3 foods-11-03981-t003:** Median (P5, P95) vitamin D intake (µg/day) for different age (years) and sex (M = male, F = female) groups of the population at background intake and optimized scenarios.

			Scenarios							
Sex	Age Group (*n*)	Background Intake	1	2	3	4	5	6	7	8
M	4–10 (*n* = 251)	2.3	11.9 (8, 19.2)	11.7 (7.8, 20.6)	12.2 (8.2, 19.7)	12.2 (8.1, 19.8)	12.1 (8.1, 19.6)	12.2 (8.1, 19.4)	12.2 (8, 19.9)	12.2 (7.9, 19.8)
M	11–17 (*n* = 216)	2.5	13.1 (8, 22.9)	13.3 (7.7, 22.9)	12.8 (7.5, 22.7)	12.9 (7.3, 23.2)	12.9 (7.4, 22.4)	13 (7.3, 23)	12.9 (7.2, 22.5)	12.9 (7.1, 23)
M	18–69 (*n* = 1329)	3.5	15.5 (8.7, 28.9)	14.2 (7.2, 27.4)	14.5 (7.7, 28.6)	14.8 (7.7, 28.6)	13.9 (7.3, 27.4)	14 (7.4, 27.5)	13.8 (7.2, 27.1)	13.8 (7.2, 27.3)
M	70–75 (*n* = 135)	4.5	17.4 (8.1, 28.4)	15.9 (7, 26.3)	15.9 (7.3, 28.3)	16.4 (7.6, 28.8)	15.2 (7.3, 25.9)	15.6 (7.2, 26.6)	15 (7.3, 25.5)	15.4 (7.2, 25.9)
M	all ages (*n* = 1931)	3.2	14.7 (8.4, 27.7)	13.8 (7.3, 26.2)	13.9 (7.7, 27.3)	14.1 (7.7, 27.7)	13.6 (7.5, 26.5)	13.7 (7.4, 26.7)	13.4 (7.4, 26.2)	13.5 (7.3, 26.3)
F	4–10 (*n* = 248)	2.2	11 (7.6, 15.6)	10.9 (7.4, 16.3)	11.2 (7.4, 16)	11.3 (7.4, 16.4)	11.1 (7.5, 16.4)	11 (7.5, 16.7)	10.9 (7.5, 16.5)	10.9 (7.4, 16.8)
F	11–17 (*n* = 215)	1.9	10 (5.6, 17.2)	10.6 (5.4, 18.8)	9.7 (5.4, 17.3)	9.7 (5.5, 17.3)	10 (5.4, 17.9)	9.9 (5.4, 17.7)	10 (5.4, 17.8)	9.8 (5.2, 17.8)
F	18–69 (*n* = 1421)	3	12.5 (7.1, 22.8)	11.3 (6, 21.7)	11.6 (6, 21.9)	11.8 (6.1, 22.1)	11.1 (5.9, 21.5)	11.2 (5.9, 21.6)	11 (5.8, 21.3)	11 (5.7, 21.4)
F	70–75 (*n* = 131)	3.6	13.2 (7.6, 23.4)	11.7 (6.1, 20.7)	12.2 (6.5, 21.4)	12.5 (6.8, 22.1)	11.8 (6.5, 19.7)	11.9 (6.6, 20.2)	11.6 (6.4, 19.3)	11.7 (6.4, 19.4)
F	all ages (*n* = 2015)	2.7	12 (7, 22.1)	11.2 (6, 20.9)	11.4 (6.2, 20.8)	11.5 (6.2, 21.3)	11 (6, 20.5)	11.1 (6, 20.6)	10.9 (6, 20.2)	10.9 (5.9, 20.3)
M, F	4–10 (*n* = 499)	2.2	11.4 (7.7, 17.7)	11.2 (7.5, 18)	11.7 (7.8, 18.6)	11.7 (7.5, 18.2)	11.6 (7.8, 18.7)	11.7 (7.7, 18.5)	11.5 (7.7, 18.6)	11.6 (7.6, 18.2)
M, F	11–17 (*n* = 431)	2.2	11.5 (6, 20.7)	11.5 (5.9, 20.3)	11.3 (6.2, 20.7)	11.4 (6.1, 20.5)	11.3 (6.1, 20.6)	11.3 (5.9, 20.5)	11.3 (6, 20.5)	11.3 (5.7, 20.5)
M, F	18–69 (*n* = 2750)	3.2	13.8 (7.7, 26.4)	12.6 (6.4, 24.8)	12.9 (6.6, 25.5)	13.1 (6.8, 26)	12.4 (6.4, 24.6)	12.5 (6.5, 24.9)	12.3 (6.3, 24.6)	12.3 (6.3, 24.5)
M, F	70–75 (*n* = 266)	4.1	14.7 (7.7, 27.4)	13.2 (6.4, 24.3)	13.7 (7.1, 26)	14 (7.1, 26.2)	13 (6.7, 25.2)	13 (6.8, 25.2)	12.8 (6.6, 24.8)	12.7 (6.5, 24.8)
M, F	all ages (*n* = 3946)	3	13.2 (7.5, 25.1)	12.3 (6.5, 23.7)	12.5 (6.7, 24.1)	12.7 (6.8, 24.6)	12.2 (6.6, 23.7)	12.2 (6.5, 23.8)	12.1 (6.4, 23.4)	12.1 (6.4, 23.5)

## Data Availability

The individual intake data are not shareable due to restrictions of use; aggregated intake data are available in the Excel workbook containing the optimization model (Appendix A). The food-composition data used for this study is publicly available at https://frida.fooddata.dk/download (accessed 15 October 2022).

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
