# Peer review of "A Data Driven Approach to Identify Safe and Adequate Schemes for Vitamin D Fortification"

_foods, 2022, doi:10.3390/foods11243981_

Round 1

Reviewer 1 Report

The reviewer appreciate the interest of the authors in the development of this manuscript. It is an interesting topic.

The aim of this study was to develop a simple and rapid automated tool to test different strategies for fortification of foods with vitamin D using intake data from a nationally representative dietary survey and taking into account the energy contribution of fortified foods.

Line: 58-59. In which studies? Please this statement needs a bibliographic reference.

Line: 134-138. Does the description apply to Figures 1 and 2? If so: there is no reference in the text to the figures and the figures should be closer to the text in which they are cited.

Tables 2 and 3 look as if they have been cut off. Please adjust the table size to fit the page.

The reviewer appreciate the interest of the authors in the development of this manuscript. It is an interesting topic. However, some indicated changes should be included in the manuscript. I suggest MINOR REVISIONS.

Reviewer 2 Report

In this manuscript, the authors address the problematic discrepancy between actual intake and recommenced intake of vitamin D. The projected results show that fortification could be an effective way to increase intake of vitamin D, in both sexes and in all age groups in Denmark. The results are very interesting and relevant, and I have some comments that I hope the authors will consider.

Abstract

·       The results do not present the results – which food groups were fortified in these projections and what was the impact on vitamin D intake?

Introduction

·       What is the current fortification in Denmark? Is there any mandatory fortification? Which foods are currently fortified (mandatory or voluntary) and if voluntary; how widespread is the fortification currently?

Methods

·       The authors do not state the measure of performance for the optimization models. Was it to increase vitamin D intake or to reach AR, RI but not UL (while keeping food intake constant)?

·       Some of the modeled scenarios included fortification of food items that are not possible to fortify. This is stated by the authors in the introduction section. Could the authors provide a rationale as to why these food groups are (e.g., fruit, vegetable, meat, fish) included?

·       What was the rationale behind setting a constriction on fortified vitamin D µg/10 mJ?

·       It is not clear if median intakes of the group or the individuals were used in the models.

Results

·       As we are currently transitioning towards a more plant-based diet in order to be more sustainable, it would be interesting to see the results in subgroups who do not consume some of the food groups (e.g., milk or cheese). How would the scenarios work in these groups?

·       Similarly, those with extremely high intake of some food groups (eg milk, cheese, oils) may

·       The last section of the results (titled “testing the scenarios”) appears to be a methodological description that belongs in the methods section.

·       No results are presented for the proportion who reached AR, RI or UL, as is stated in the aim.

Discussion:

·       In the first section, the reference to dansda should be corrected.

·       Here, it is clear that RI-UL were the limits the models were targeting. This should be clear in the methods section.

·       The authors should discuss the implications of these results in terms of public health and sustainability (from a ecological perspective). While it is realistic to assume intake of foods remain constant, food choices should probably be shifted towards healthier and more sustainable options. What implications would this have on the projected intakes (i.e., if animal foods or saturated fats are consumed less, and plant foods or unsaturated fats are consumed more)?

·       The conclusion should present the main results of the study.

References:

·       There are other studies that modelled fortification scenarios, that are not mentioned in the text (Bruins, M.J.; Létinois, U. Adequate Vitamin D Intake Cannot Be Achieved within Carbon Emission Limits Unless Food Is Fortified: A Simulation Study. Nutrients 2021, 13, 592.). Since there are few references cited overall, this reference (and possibly others I am unaware of) would be a nice addition to the introduction or discussion.

Figures and tables:

·       Table is a bit difficult to read due to the different alignments of the column headings and the data within the columns. Using the same alignment for both column heading and the data would be helpful.

·       Figure 1-2: It is a bit difficult to read the text below the boxplot (referring to the scenarios). It could be sufficient to only state the scenario number (e.g., 1, 2, 3 etc) and describe the differences in a footnote.

Round 2

Reviewer 2 Report

I congratulate the authors on a successful revision. I have no further comments.